# Manipulation, Sampling and Inactivation of the SARS-CoV-2 Virus Using Nonuniform Electric Fields on Micro-Fabricated Platforms: A Review

**DOI:** 10.3390/mi14020345

**Published:** 2023-01-29

**Authors:** Devashish Mantri, Luutzen Wymenga, Jan van Turnhout, Henk van Zeijl, Guoqi Zhang

**Affiliations:** 1Department Biomedical Engineering, Delft University of Technology, 2628 CD Delft, The Netherlands; 2Department Microelectronics, Delft University of Technology, 2628 CD Delft, The Netherlands; 3Department Material Science Engineering, Delft University of Technology, 2628 CD Delft, The Netherlands

**Keywords:** micro-electrodes, virus in-activation, virus sampling, virus concentration, SARS-CoV-2, dielectrophoresis

## Abstract

Micro-devices that use electric fields to trap, analyze and inactivate micro-organisms vary in concept, design and application. The application of electric fields to manipulate and inactivate bacteria and single-celled organisms has been described extensively in the literature. By contrast, the effect of such fields on viruses is not well understood. This review explores the possibility of using existing methods for manipulating and inactivating larger viruses and bacteria, for smaller viruses, such as SARS-CoV-2. It also provides an overview of the theoretical background. The findings may be used to implement new ideas and frame experimental parameters that optimize the manipulation, sampling and inactivation of SARS-CoV-2 electrically.

## 1. Introduction

Viruses, often debated to be non-living bodies, are complicated assemblies of proteins, nucleic acids and carbohydrates. Their incredibly small size (17–400 nm) makes it a challenge to manipulate them. Research on viruses becomes more urgent as health risks posed by viral pathogens become more frequent and deadly. Severe Acute Respiratory Syndrome Corona Virus 2 or SARS-CoV-2, a virus belonging to the corona virus family and the microbe responsible for the COVID-19 pandemic, has claimed over 6.7 million lives across the globe, as of December 2022 [1]. Mitigation of the risks posed by such lethal viruses can be brought about when we are able to successfully analyze the virus. A thorough review about the prevention, diagnosis and treatment of COVID-19 is given by Aliabadi et al. [2]. They also discuss the opportunities offered by nanoscience and nanotechnology.

Studying the mechanisms of virus deactivation could have huge benefits for battling pandemics such as COVID-19. Until now, various chemical [3], optical [4,5] and thermal methods [6] have been developed to perform these tasks. However, most of the techniques using chemical or optical agents to trap, count and inactivate the virus prove to be cumbersome, and are difficult to integrate onto a single platform. Optical methods to quantify viruses often need expensive equipment and well-trained operators [2,7]. Analysis and deactivation using chemical microbial agents are time consuming, as they need sample preparation and are not label free [7].

Nonuniform electric fields have been used before, albeit separately, to trap, analyze and inactivate single-celled organisms and bacteria. Viruses, on the other hand, are more difficult to manipulate and show resistance to inactivation treatments using electric fields as high as 2.9 × 10^6^ V/m [8] due to their small size. Micro-electrodes, engineered using micro-fabrication technology, open doors to an all-in-one integrated system to perform all tasks simultaneously. New developments in micro-fabrication allow us to scale down and create high electric field regions between the electrodes, even with moderate power. These new developments in micro-electronics incite renewed interest in the lysis/inactivation of viruses on micro-platforms and can be used to develop viable products, such as face masks, laboratory equipment (lab on chip devices (LOC) and micro-total analysis systems (μTAS)) for low-cost sampling and instant inactivation of the virus. Studies have focused, for instance, on triboelectric nanogenerator (TENG) masks to charge electret fibers, or on the use of quantum dots, to study the effect that electrostatics and electric fields generated by the fabric and the dots have on the filtration and inactivation of micro-organisms [9,10]. Similarly, electric filtration and sampling techniques on LOC devices are being explored [11].

Previous reviews have only addressed the concentration of viruses using electric fields [12]. This review explores the techniques used to capture, assess and inactivate viruses electrically, and discusses their effect, especially on the SARS-CoV-2 virus. As stated earlier, limited research has been performed on the manipulation and killing of smaller viruses. Hence, this review aims to survey the electrical quantification methods and inactivation mechanisms of viruses, and to conceive an experimental setup for the manipulation and inactivation of the SARS-CoV-2 virus. The review is arranged as follows: First, we examine techniques to concentrate the virus in a given liquid medium on planar electrodes. Next, methods to quantify the virus are investigated and virus inactivation methods are described. Finally, electrode geometries that can optimize concentration, quantification and inactivation on a single platform are dealt with.

## 2. Virus Manipulation Using Electric Fields

### 2.1. Concentration of Moderately Large Viruses (>200 nm)

A polarizable particle suspended in a liquid medium exposed to a nonuniform electric field experiences a force known as the dielectrophoretic force (DEP). When the medium is more polarizable than the particle, the particle will be repelled by the electrode, resulting in a negative DEP force (nDEP). Whereas when the particle is more polarizable, than the medium, the particle, will be attracted to the electrode, resulting in a positive DEP force (pDEP). Dielectrophoretic manipulation has been used to successfully trap and concentrate bacteria and viruses. Concentration of the viral particulates facilitates the formation of localized clusters that can be easily examined and inactivated. Morgan et al. successfully demonstrated the use of DEP in concentrating viruses [13]. They concentrated Tobacco Mosaic Virus (TMV) (ϕ 280 nm) with a pDEP using saw-tooth electrodes with 2–6 μm pitch spacing. The virus was found to be highly polarizable, which was attributed to the absence of an insulating membrane. The group also showed that TMV could be separated and filtered out from a mixture of TMV and Herpes Simplex Virus (HSV, ϕ 150–240 nm) by exploiting the difference in the Clausius–Mossotti (CM) factor. The CM factor depends on the virus’s composition and structure [14]. Finally, the group explored the prospects of DEP on HSV. The HSV was concentrated by a pDEP force with an electric field (freq. 4.5 MHz, 5 V_pp_, E = 10^6^ V/m) [15]. The manipulation of the Vaccinia virus (ϕ 240 nm) is also well documented. A study reported the use of interdigitated Ti/Pt electrodes to trap Vaccinia virus using a pDEP with 7 V_pp_ and 1 MHz [16].

Recently, the use of nanofibers for trapping viruses was investigated [17]. Although, the nanofibers are particularly good at trapping the virus from a flowing sample, they cannot concentrate the virus. Similar inferences can be gathered from studies that suggests the use of electrostatics to trap the corona virus [18,19]. The virus can only be trapped, but not concentrated. The viruses described above are moderately large DNA viruses with an inner surface area high enough to amass enough charges to facilitate its manipulation. As particle dimensions shrink, the controlled manipulation of particles becomes increasingly difficult as induced dipole moments scale with the third power of the particle’s radius [12]. This could be partly compensated by increasing the electric field strength through electrode geometry optimization and/or increasing the source voltage. However, the latter likely results in the electrolysis of the medium.

### 2.2. Concentration of Smaller Viruses

Smaller viruses were also shown to respond to the dielectrophoretic effect. Influenza (ϕ 80 nm) and Hepatitis virus (ϕ 32 nm) were manipulated and concentrated using negative dielectrophoresis (nDEP) on planar electrode arrays. Planar platforms and 3D structures that make use of quadrupole and octupole electrode cages establish points of confluence, where the virus can be trapped effectively [20]. Cowpea Mosaic Virus (CPMV) (30 nm), a spherical virus, was successfully trapped using castellated electrodes with a pitch size of 2 μm [21]. Despite having a small size, CPMV shows a higher polarizability than that of the medium, because of its non-enveloped nature. The same is true for the Adenovirus and Rotavirus, which despite having a small size, were easily trapped and concentrated using a pDEP [22]. The pDEP trapping of viruses was observed in media with higher solution conductivities than that of bacteria. The prime reason for this might be attributed to the smaller size of viruses compared to that of bacteria.

A challenging problem is that with the increase in electrolyte conductivity, the ionic strength increases. This may result in more electrochemical reactions. Usually, virus samples require a relatively high ionic strength for their storage. Another challenge is that smaller viruses (<100 nm) show a random Brownian motion. Brownian motion is characterized by random movements that could hinder the trapping of viruses [22,23]. These problems can be resolved by attaching the virus to a larger molecule (shown in Figure 1), e.g., Hepatitis A virus (27 nm) bonded to a streptavidin-coated particle [24].

Insulator-based DEP (iDEP) is a technique that uses insulating structures to concentrate the electric field. The technique seems to become more common for concentrating smaller viruses as well as larger viruses in recent years. For example, influenza virus (90 nm) [25], Sindbis virus (130 nm) [26], bacteriophages such as T4 and SPN3UP (90 nm) [27] and other larger viruses such as TMV [28] have been concentrated by iDEP. A study using circular and oval-shaped electrodes employed this method and was devised to trap three different strains of the same bacteriophage, demonstrating the high specificity of this technique [27]. However, no study has established a better efficiency of iDEP over DEP in terms of trapping and concentrating the smaller virus particles. Moreover, iDEP devices are more susceptible to Joule heating and electrolysis of the medium [29].

A summary of all the different aspects of the practical studies that concentrated viruses using DEP have been tabulated in Table 1 below.

### 2.3. Trapping of SARS-CoV-2: Theoretical Approach

The spherical SARS-CoV-2 virus is characterized by a lipid insulating membrane, and hence, the particle may not be easily polarizable. Tuning properties of the medium becomes instrumental if we desire to attract and concentrate the virus particles with pDEP. The Clausius–Mossotti factor of SARS-CoV-2 can be modelled by using a core-shell model, as depicted in Figure 2. The inner layer represents the core and the outer layer represents the lipid membrane. The DEP force on a neutral particle can be described by:(1)F=2πεoεs′R3Re[(εp*−εs*)/(εp*+2εs*)]∇E2
where $\varepsilon_{p}^{*}$ is the permittivity of the particles, $\varepsilon_{s}^{*}$ that of the suspending solvent, R the radius of the particle, E the applied field, and ∇ the divergence. According to Equation (1), the force can be attractive or repulsive depending on the values for $\varepsilon_{p}^{*}$ and $\varepsilon_{s}^{*}$. The ratio of permittivities in Equation (1) equals the CM-factor, so:(2)$$CM^{*}=\left(\varepsilon_{p}^{*}-\varepsilon_{s}^{*}\right)/\left(\varepsilon_{p}^{*}+2\varepsilon_{s}^{*}\right)$$

Note that the permittivities are complex quantities, the real and imaginary part of which usually will change with the applied radial frequency *ω*. As a result, *F* will depend on *ω* as well. The fact that $\varepsilon_{p}^{*}$ and $\varepsilon_{s}^{*}$ and, thus, *CM** are complex is indicated in Equations (1) and (2) by the asterisk in superscript. The symbol $\varepsilon_{s}^{\prime }$ means the real part of $\varepsilon_{s}^{*}$. The imaginary part is denoted by $\varepsilon_{s}^{\prime \prime }$. We may thus write:(3)$$\varepsilon_{s}^{*}(\omega )=\varepsilon_{s}^{\prime }(\omega )-i\varepsilon_{s}^{\prime \prime }(\omega )$$

If $\varepsilon_{s}^{\prime \prime }$ is caused by conduction losses, then $\varepsilon_{s}^{\prime \prime }\left(\omega \right)=\gamma_{s}/\left(\varepsilon_{o}\omega \right)$ where *γ_s_* is the conductivity of the solvent and *ε_o_* the permittivity of vacuum *ε_o_* = 8.854 × 10^−12^ F/m. In addition to ionic conduction, dipole relaxations may also occur in a dielectric medium. We assume that these are not strong in the present case and so they are neglected. This implies that $\varepsilon_{s}^{\prime }$ will remain constant. We will make the same assumptions for the complex permittivities of the core and the membrane of the virus, $\varepsilon_{c}^{*}$ and $\varepsilon_{m}^{*}$. This means that:(4)$$\varepsilon_{s}^{*}(\omega )=\varepsilon_{s}^{\prime }-i\gamma_{s}/(\varepsilon_{o}\omega ),\;\varepsilon_{c}^{*}(\omega )=\varepsilon_{c}^{\prime }-i\gamma_{c}/(\varepsilon_{o}\omega ),\;\varepsilon_{m}^{*}(\omega )={\varepsilon^{\prime }}_{m}-i\gamma_{m}/(\varepsilon_{o}\omega )$$

The change in the CM-factor with frequency *f*, *f* = *ω*/(2π), will evidently derive from the conductivity and permittivity of the solvent, as well as from the conductivity and permittivity of the core and membrane of the particle.

For a layered sphere, the equation for $\varepsilon_{p}^{*}$ obeys:(5)$$\varepsilon_{p}^{*}=\varepsilon_{m}^{*}\frac{(R/r{)}^{3}+2\left(\varepsilon_{c}^{*}-\varepsilon_{m}^{*}\right)/\left(\varepsilon_{c}^{*}+2\varepsilon_{m}^{*}\right)}{(R/r{)}^{3}-\left(\varepsilon_{c}^{*}-\varepsilon_{m}^{*}\right)/\left(\varepsilon_{c}^{*}+2\varepsilon_{m}^{*}\right)}$$

which may also be written as:(6)$$\varepsilon_{p}^{*}(\omega )=\varepsilon_{m}^{*}(\omega )\frac{2(R^{3}-r^{3})\varepsilon_{m}^{*}(\omega )-(R^{3}+2r^{3})\varepsilon_{c}^{*}(\omega )}{\left(2R^{3}+r^{3})\varepsilon_{m}^{*}(\omega )+(R^{3}-r^{3})\varepsilon_{c}^{*}(\omega \right)}$$

The core-shell structure of the virus is visualized in Figure 2, in which the permittivity of the RNA-core $\varepsilon_{c}^{*}$ and that of the membrane proteins $\varepsilon_{m}^{*}$ are complex, frequency dependent quantities. Both are assumed to obey the expressions given in Equation (4). The radius of the core is *r* and that of the outer layer is *R*. When we substitute Equation (5) in the CM factor given by Equation (2), we obtain:(7)$$\frac{\varepsilon_{p}^{*}(\omega )-\varepsilon_{s}^{*}(\omega )}{\varepsilon_{p}^{*}(\omega )+2\varepsilon_{s}^{*}(\omega )}=\frac{R^{3}\left(\varepsilon_{c}^{*}(\omega )+2\varepsilon_{m}^{*}(\omega )\right)\left(\varepsilon_{m}^{*}(\omega )-\varepsilon_{s}^{*}(\omega )\right)+r^{3}\left(\varepsilon_{c}^{*}(\omega )-\varepsilon_{m}^{*}(\omega )\right)\left(2\varepsilon_{m}^{*}(\omega )+\varepsilon_{s}^{*}(\omega )\right)}{R^{3}\left(\varepsilon_{c}^{*}(\omega )+2\varepsilon_{m}^{*}(\omega )\right)\left(\varepsilon_{m}^{*}(\omega )+2\varepsilon_{s}^{*}(\omega )\right)+2r^{3}\left(\varepsilon_{c}^{*}(\omega )-\varepsilon_{m}^{*}(\omega )\right)\left(\varepsilon_{m}^{*}(\omega )-\varepsilon_{s}^{*}(\omega )\right)}$$

As indicated, the *CM** given by Equation (7) will depend on the frequency. The same applies to the DEP force given by Equation (1), as we have already mentioned above.

We should realize that $\varepsilon_{s}^{*}$ from the suspending solvent will differ from the permittivity of the solvent proper. The solvent with virus particles forms a suspension, a mixture. For the calculation of the permittivity of a mixture, many formulae are available. A convenient one of Maxwell-Garnett, based on the mean field approximation, gives an explicit formula for the permittivity of a suspension of spherical particles:(8)$$\varepsilon_{sp}^{*}(\omega ,v)=\varepsilon_{s}^{*}(\omega )\frac{2(1-v)\varepsilon_{s}^{*}(\omega )+(1+2v)\varepsilon_{p}^{*}(\omega )}{\left(2+v)\varepsilon_{s}^{*}(\omega )+(1-v)\varepsilon_{p}^{*}(\omega \right)}$$

Note that $\varepsilon_{sp}^{*}$ may also written as:(9)$$\varepsilon_{sp}^{*}(\omega ,v)=\varepsilon_{s}^{*}(\omega )\frac{\varepsilon_{p}^{*}(\omega )+2\varepsilon_{s}^{*}(\omega )+2v[\varepsilon_{p}^{*}(\omega )-\varepsilon_{s}^{*}(\omega )]}{\varepsilon_{p}^{*}(\omega )+2\varepsilon_{s}^{*}(\omega )-v[\varepsilon_{p}^{*}(\omega )-\varepsilon_{s}^{*}(\omega )]}$$
where *v* is the volume fraction of the virus particles. It is this $\varepsilon_{sp}^{*}$ which we should substitute for $\varepsilon_{s}^{*}$ in Equations (2) and (7). This leads to:(10)$$CM^{*}(\omega ,v)=\frac{\varepsilon_{p}^{*}(\omega )-\varepsilon_{sp}^{*}(\omega ,v)}{\varepsilon_{p}^{*}(\omega )+2\varepsilon_{sp}^{*}(\omega ,v)}$$

We can compute *CM* from Equations (5), (8) and (9) with a few lines of code using Matlab, Maple or Mathematica. The plots shown in Figure 3 are obtained with a Mathematica script. Details about the script are provided in the Appendix A.

Selecting a non-aqueous medium, such as that of *γ_s_* < *γ_p_*, a pDEP force on the corona virus can be produced at lower frequencies to effectively concentrate the virus. Unfortunately, studies have not delineated the electrical parameters of SARS-CoV-2. Nonetheless, for a KCl solution with a low conductivity, and as long as $\varepsilon_{s}^{\prime }>\varepsilon_{c}^{\prime },\varepsilon_{s}^{\prime }>{\varepsilon^{\prime }}_{m}$, the graph in Figure 3 shows that the virus will be attracted to the electrodes at acceptable frequencies, for which the real part of *CM* becomes positive. Figure 3 also suggests that with a lower medium conductivity, pDEP can be brought about by lower frequency electric fields. This calculation does not consider the negative charges that are already present on the RNA molecule inside the core of the virus. This assumption is supported by previous studies that show that dielectrophoretic mobility is not affected by the interior charges in Cowpea Chlorotic Mottle virus [30] as well as the corona virus [18]. If $\varepsilon_{s}^{\prime }>\varepsilon_{c}^{\prime }$ and a medium conductivity of <0.005 S/m is maintained, there is a high chance that corona virus can be concentrated with moderately high frequencies and electric fields.

## 3. Electric Sampling

Micro-electrodes can be employed as sensors to quantify the virus present in a sample. The use of electric fields to count viruses dismisses the need for tagging and sample preparation. The usual method used for detecting and quantifying a specific virus is the Polymerase Chain Reaction (PCR), which is highly accurate, but not rapid. Significant work has been carried out on viruses where differences in the electrical parameters between the particle and the medium are utilized to give information about the quantity and type of virus present. Other techniques used to detect the presence and type of virus use antibodies coated on nanowires that record a change in conductance when a virus particle binds to the antibody [31]. This method provides a higher selectivity and faster simultaneous detection than the PCR method.

Dielectrophoretic impedance measurement (DEPIM) is a technique that can quantify the trapped virus without the need for antibodies. The DEPIM technique is a combination of DEP manipulation and impedance measurement, which records the change in impedance as soon as a virus is trapped (Figure 4). Numerous studies have demonstrated the use of DEPIM to quantify bacteria [32,33] and yeast cells [34]. Nakano et al. showed, for the first time, that DEPIM can also be applied to obtain information about the quantity of the virus in a solution. This research group has been able to detect and sample Adenovirus and Rotavirus [22]. Similar research has been performed to sample and quantify Vaccinia virus using DEPIM [35]. DEPIM demonstrates dual functionality for a device as it allows trapping as well as sampling. The DEP behavior of the Adenovirus and the Rotavirus was observed by varying the electrical conductivity of the suspension to find out the effective trapping value. When trapped with a voltage of 5 V at 100 kHz, the change in the conductance was monitored. Similarly, for Vaccinia virus, 8 V with a 1 kHz frequency was used and the detection reached a number as low as 2.58 × 10^3^ particles/mL. This technique can be easily integrated onto platforms that are designed to trap and inactivate viruses.

Another technique used to quantify and sample viruses treats the virus as an impurity and then measures the change in capacitance between the electrodes. The measured change in capacitance, as a result of the difference in permittivity of the virus particulate and the conducting medium, can then be empirically traced to the number of particles present in the medium by applying bioelectrochemical theory [36]. The electrodes in this study were established as resonating 3D chambers. Nonetheless, the same theory could be extended to planar electrodes, or simple 3D electrodes, and could also be integrated with trapping and killing platforms. Many research endeavors have also highlighted that if we plot the medium conductivity vs. the mobility of the virus, we obtain a graph that is unique for each kind of virus. Similarly, unique data points can be established for a given virus if we plot the phase of the imaginary impedance at a frequency where the impedance is at a maximum against an arbitrary frequency. These techniques can be applied to identifying the virus type [37,38].

This method utilizes the specificity of the virus’s composition, which is reflected by the effective permittivity of the membrane to identify the virus. An extension of the same method would be to measure the electrical capacitance per virus particle to find out the specific dielectric constant, and then subtract the contributions of the medium from the measured capacitance to identify the virus [23]. The capacitance of the suspension with virus particles is modelled, as shown, in Figure 5, viz. as the sum of two parallel capacitances, one filled with the suspending medium and the other with all virus particles.

Another approach uses co-planar differential capacitive sensors (refer to Figure 6). Two capacitive sensors, as part of a differential capacitor, are loaded with the medium and the sample solution, respectively. The differential capacitor, due to the deposited nanoparticles, is modelled as an effective homogeneous medium. The study suggests that, with an appropriate effective medium theory, the effective dielectric constant can be related to the number and composition of the viruses [39]. The study investigates nanoparticles, but the theory can be extended to virus particles as well.

However, the calculations depend on the evaporation times of the media and a fast integrated analysis within the device itself may be a challenging goal.

A list of practical studies that successfully sampled viruses using electric fields has been tabulated in Table 2, given below.

## 4. Electrical Inactivation

The electrical inactivation of bacteria or viruses is a technique that refers to a process that invokes electric fields to render the organism incapable of replicating or infecting a host. Ions generated by electric activities, such as the corona discharge, have been proven to be effective in inactivating the virus [40,41]. This inactivation involves the creation of highly reactive radicals that bring about changes to the protein structure. Electrical inactivation can be fast and direct. Electrical inactivation can be brought about by electrical lysis, which refers to the breakdown of the cell membrane/membrane proteins. In the past 20 years, in response to the increased potency and deadliness of airborne viruses, macro-scale units to trap and kill viruses have been developed. These macro-scale units are either in the form of filters [42], electrified face masks [43,44], or particle concentrators [45,46]. Micro-devices designed for electrical inactivation have been utilized for bacteria and single-cell organisms. Very rarely do the studies extend to the inactivation of viruses. To construct a viable device that can kill the SARS-CoV-2, we need to review the mechanisms used to inactivate smaller bacteria and larger viruses.

### 4.1. Inactivation by Irreversible Electroporation

The electrical inactivation of bacteria is usually brought about by electrical lysis and irreversible electroporation. The lysis results from the creation of a transmembrane potential that is directed from the outside to the inside of the cell as a result of the accumulation of charges. When the transmembrane potential exceeds a certain threshold, electroporation occurs. With a further increase in the electric field, the electroporation is made permanent, which results in lysis. The transmembrane voltage to achieve cell lysis is around 1 V [47], which requires a field of about 10^7^ V/m. Lysis by irreversible electroporation demands a conductive medium. A higher conductivity of the medium leads to higher irreversible electroporation and, thus, a higher lysis rate [48]. This results in an interesting trade-off, whereby increasing the conductivity of the medium may foster higher killing rates but will lower the concentration efficiency. This results, eventually, in a lower total inactivation of the virus. The medium conductivity needs to be specifically tailored to optimize both the concentration and inactivation. Furthermore, the parameters used for the electrical excitation are also crucial in achieving higher inactivation rates.

Many studies used pulsed excitation to kill bacteria [49,50,51]. With pulsed electric fields, the parameters that need to be set are the number, shape, amplitude and width of the pulse. The parameters used across lysis experiments are influenced by the shape and size of the particle being lysed. However, in general, in bacteria as well as in cells, it was noted that higher lysis levels were attained with a longer pulse duration, while higher pulse amplitudes were needed to lyse particles with smaller diameters [52,53,54]. If a medium with a low conductivity is used, a higher pulse amplitude and a larger pulse duration is required to achieve irreversible electroporation [55]. Higher lysis rates were observed with bipolar rectangular pulses than with sinusoidal pulses [56]. Another study maintains that monopolar pulses were found to be better than bipolar pulses [53]. Although a larger pulse amplitude may facilitate a higher lysis rate, it may also cause electrolysis of the conducting medium. To avoid electrolysis, an AC excitation can be employed. Cells and bacteria have been effectively lysed using AC [57,58,59]. In general, while the large magnitude of ∇E^2^ is useful to induce a strong DEP force in most dielectrophoretic studies, an increase in the field strength E leads to electroporation. We can optimize the geometry such that the divergence as well as the field strength are maximized to attract the particles at points of high electric field for an effective lysis.

The only paper that claims the inactivation of a virus by irreversible electroporation used AC to concentrate and lyse the virus. The study reported the electrical lysis of Vaccinia virus [38]. The damaged virus particles and DNA traces tagged by fluorescent agents were found on the chip surface and confirmed with SEM images (refer to Figure 7).

Similarly, a study in 2014 reported, although inadvertently, the lysis of Vaccinia virus on nanofiber probe arrays [35]. Both studies shed some light on the electrical lysis of viruses and the mechanisms used to bring about irreversible electroporation. However, no other study discusses electrical lysis of smaller RNA viruses. It could be that the voltage needed to establish a lethal transmembrane potential across a smaller virus is practically unfeasible, or that the surface properties of smaller viruses do not allow the development of a lethal transmembrane potential. Either way, very little research has been conducted on the effect of electric fields on the inactivation of smaller viruses by irreversible electroporation. A list of practical studies that successfully inactivated biological samples using irreversible electroporation has been compiled in Table 3, given below.

#### Irreversible Electroporation of SARS-CoV-2

As iterated before, the membrane voltage to lyse cells is around 1 V. Ignoring the spikes, we may model SARS-CoV-2 as a core with one shell. The core-shell model is redrawn in Figure 8. We can assume the virus to be suspended in a buffer solution with a relatively high (ionic) conductivity $\gamma_{e}$. The core of the virus will most likely also have a significant conductivity $\gamma_{c}$. The membrane, which consists of lipids, can be expected to have a low conductivity $\gamma_{m}$. These parameters of the virus were hinted at by Sholanov in his scientific essay [60]. The conductivities rule the steady state value of the membrane voltage if a DC step voltage is applied. This value can be derived by solving the Laplace equation for a layered spherical particle. A comprehensive description of the derivation can be found in Kotnik et al. [55,61]. They derived the following expression:(11)$$V_{m}=fE_{a}R\cos\theta$$
where *f* is the conductivity factor, *E_a_* the applied field, *R* the radius of the virus, and *θ* the direction of *E_a_*. The factor *f* equals:
(12)$$f=\frac{2\gamma_{e}\left({\left(\gamma_{m}-\gamma_{c}\right)}^{3}(r/R{)}^{3}+\gamma_{c}-3\gamma_{m}r/R+2\gamma_{m}\right)}{2(\gamma_{c}-\gamma_{m})(\gamma_{m}-\gamma_{e})(r/R{)}^{3}+(\gamma_{c}+2\gamma_{m})(\gamma_{m}+2\gamma_{e})}$$
where *r* is the radius of the core. Note that we have modified Kotnik’s formula to the one given earlier by Neumann in 1989 [47]. Schwan et al. have given an approximation of the formula for the final value of *V_m_* in response to a step voltage [62]:(13)$$V_{m}≃3/2E_{a}R\cos\theta$$
Here, the 3/2 coefficient corresponds to f when *γ_m_* = 0. To produce a transmembrane potential of 1 V we would need a field of 8.3 V/μm. Grosse et al. extended the model to a spherical cell with two layers [63]. The expression for f then reduces to Equation (12), if the thickness of the second layer is insignificantly thin. Under the influence of AC voltages, conductivities in Equations (10) and (11) are replaced by complex permittivities. This leads to an f that depends on the radial frequency *ω*.

### 4.2. Short Pulse Effect

Another lysis mechanism that evades the need for a conductive medium is the short pulse effect, which exploits the inhomogeneities in the cytoplasm of the bacterium. Pulse durations of an order smaller than the charging time of the bacterial cell membrane cause local heating near and around the inhomogeneities of the cellular material, causing cell lysis [50,52]. Generally, the electrical pulses do not affect the intercellular organelles, because the outer membrane shields the interior from the influence of electrical fields. When the pulse width becomes smaller than the charging time of the membrane, the intracellular organelles can be directly damaged without damaging the extracellular membrane. This mechanism of inactivation is widely successful for decontaminating liquids [64,65]. However, this technique has never been used to inactivate viruses. The theory describing the influence of nanosecond impulses of high voltage on the inactivation of cells reveals that the optimum pulse width depends on the relative permittivity of the cell membrane and the cytoplasm and is directly proportional to the size of the cell [66]. For small viruses, such as SARS-CoV-2, the needed pulse widths are incredibly small and reaching these small values seems infeasible.

### 4.3. Damaging the Spike Protein

In 2021, a lot of studies focused on the inactivation of SARS-CoV-2 using agents other than the primary chemical and optical disinfectants. Chief among these studies was the simulation study carried out by Arbeitman et al. [67] that suggested that the use of moderate electric fields could be sufficient to inactivate the corona virus.

Through molecular dynamic simulations, Arbeitman et al. showed that electric fields of 10^5^−10^7^ V/m stretch the dipoles found in the spike proteins of the virus and change the protein structure permanently. Irreversible structural changes especially on the receptor binding domain (RBD) of the spike protein, render the virus incapable of docking to the host cell’s receptors and thus inactivate the virus (see Figure 9).

Another study elucidated the effect of pulse electric fields on the conformation changes of the virus’ proteins [68]. It suggests that different amplitudes and intensities of pulses have different effects on the conformational changes at different locations of the RBD on the spike protein. A stimulus with a higher amplitude and higher intensity is better at reducing the structural stability of the protein. The conclusion from this study is the same as that from Arbeitman’s study: Corona virus can be deactivated using moderate electric fields. The fields used for the simulations in this study however were higher than the fields used in previous study (1.5−2.9 × 10^9^ V/m).

## 5. Electrode Configurations for Trapping, Sampling and Killing

Using electric fields allows us to integrate concentration, quantification, and inactivation on a single microchip platform. For this, choosing the electrical parameters of the conducting medium is essential in trapping the virus at points of high electric field and achieving optimum quantification and inactivation. The electrode geometry is another crucial parameter. Geometries that maximize concentration as well as killing should be employed. A variety of customized configurations have been used to concentrate and inactivate bacteria and viruses. These configurations are either 3D or 2D. Of the two, 3D electrode configurations produce a DEP force with a larger divergence and, hence, have a greater penetrating power within the sample.

For concentrating and killing smaller viruses, studies focus on 3D nano-electrode arrays with carbon nanotubes that create extremely high electric field strengths at the tips [17,35,69]. A few studies have demonstrated that micro-channels can be used to generate high electric fields by controlling the channels width. These 3D structures are shown to be more effective at concentrating bigger organisms [70,71]. Furthermore, various studies that compared the performance of 3D electrodes to that of 2D electrodes maintain the superiority of 3D electrodes in the inactivation of various larger microbes, such as bacteria and yeast cells [72,73]. Smaller viruses could also be more susceptible to inactivation fields created by 3D electrodes than by 2D electrodes. However, 3D electrodes are more difficult to fabricate and often need a microchannel to contain the sample.

Planar electrodes are very easy to fabricate; for example, interdigitated electrodes, castellated electrodes or matrix electrodes. Interdigitated or finger electrodes create a less nonuniform electric field with few high electric field points. These configurations are used mainly for dielectrophoresis [16,22,74,75]. Nonetheless, interdigitated electrodes can still be used for the lysis of larger cells and bacteria (> 1 µm) [76]. The 2D electrode layouts, such as the quadrupole [20,77,78] and micro-well electrodes [79,80], are good for single-cell analysis and do not generate many high electric field points for effective concentration and inactivation. Whereas electrode geometries, such as the matrix geometries [81,82], create many local electric field points that facilitate concentration. However, the local electric fields may not be high enough for an efficacious inactivation. In another paper, it was found that oval ducts are better at generating nonuniform fields than circular ducts [27]. An example of the lay-out of interdigitated electrodes with spikes is shown in Figure 10.

## 6. Conclusions

Using electric fields as a tool may allow us to integrate the trapping, sampling and inactivation of viruses on a single microchip platform. For this, optimizing the electrical parameters of the conducting medium is instrumental in facilitating a pDEP to trap the virus between the electrodes at points of high electric field to facilitate efficient sampling and inactivation. Sampling can be brought about by DEPIM measurements by measuring the change in capacitance between the planar electrodes right after the viruses are trapped. The electrode geometry is another sensitive parameter. Geometries that maximize both trapping and killing should be employed. The prime inactivation mechanism of SARS-CoV-2 is still unclear. Inactivation by irreversible electroporation and spike deactivation seem to be equally probable mechanisms that may take course when the virus is subjected to an electric field. Future experimental research should focus on exploring the inactivation mechanism of SARS-CoV-2 to confirm or dismiss the possibility of the inactivation of SARS-CoV-2 using moderate electric fields. Insight into the molecular interactions of the virus’s proteins and membrane lipids with electric fields and voltage potentials can help to establish a proof of concept. With a firm proof, products such as electric facemasks can be better designed, and a new generation of micro-fiber filters can be developed and deployed, which may help us ward off future pandemics.

## Figures and Tables

**Figure 1 micromachines-14-00345-f001:**
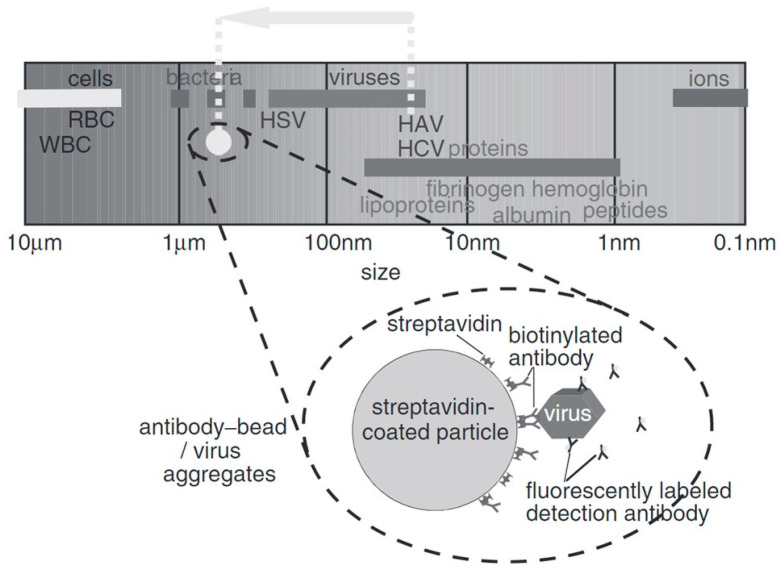
The size of a virus can be modified to facilitate manipulation by bonding it to another microsphere, from [24]. Copyright obtained from the publisher.

**Figure 2 micromachines-14-00345-f002:**
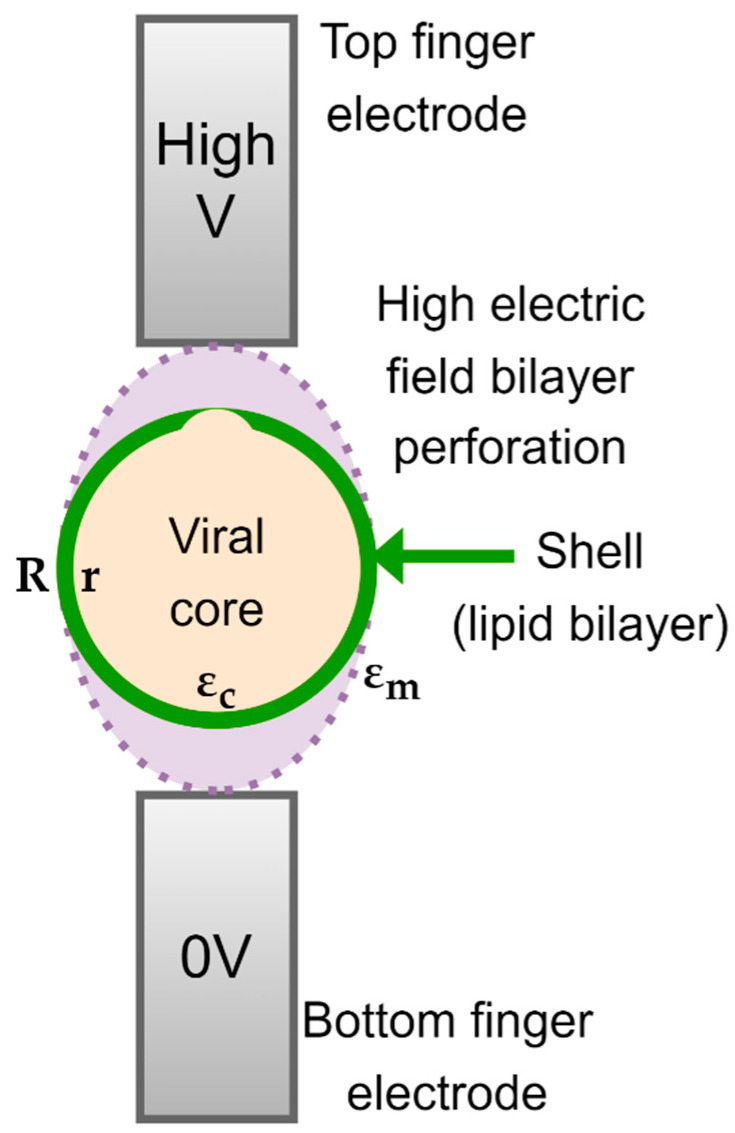
Core-shell model of SARS-CoV-2.

**Figure 3 micromachines-14-00345-f003:**
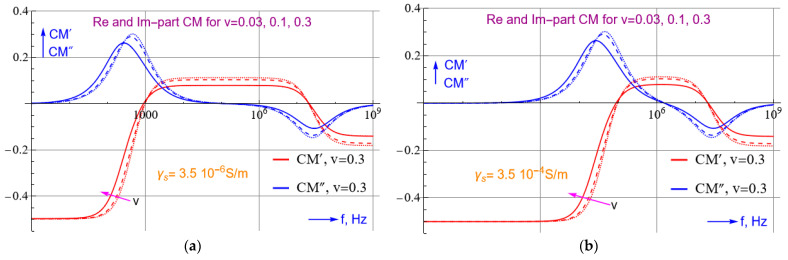
The real and imaginary CM-spectra calculated for two buffer medium conductivities of $\gamma_{s}$ = 3.5 × 10^−6^ (**a**) and 3.5 × 10^−4^ S/m (**b**) and three volume fractions v of the virus. *CM*′ becomes positive only in a certain frequency region. This region narrows if the conductivity of the buffer increases. The *CM*″ spectra can be approximated roughly by d*CM*′/dln*ω*. They thus show a peak when the slope of *CM*′ changes the most. The position of these peaks corresponds roughly to those in the dielectric loss spectra. The spectra given hold for $\varepsilon_{s}^{\prime }$ = 78, $\varepsilon_{c}^{\prime }$ = 70, $\varepsilon_{m}^{\prime }$ = 12, $\gamma_{s}$ = 3.5 × 10^−4^ S/m, $\gamma_{c}$ = 0.2 S/m, $\gamma_{m}$ = 10^−9^ S/m, *r* = 54 nm and *R* = 60 nm. The modelling was performed with Mathematica, cf. Section A.1.

**Figure 4 micromachines-14-00345-f004:**
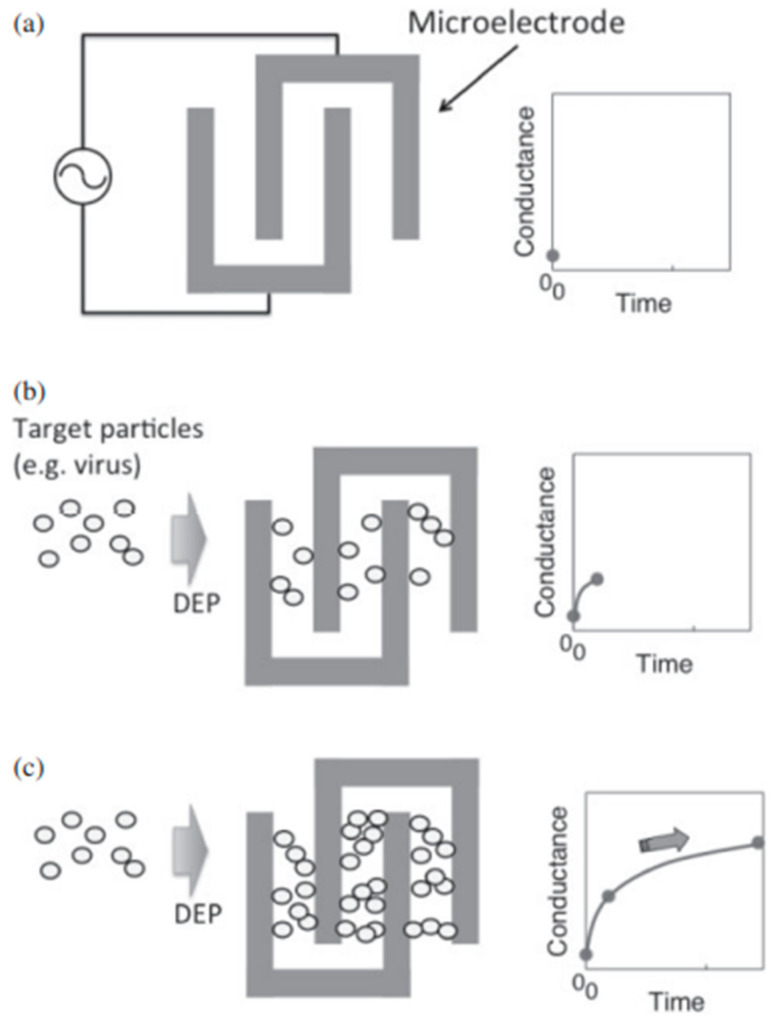
(**a**–**c**) Using DEPIM to measure and quantify virus particles, from [22]. Copyright obtained from the publisher.

**Figure 5 micromachines-14-00345-f005:**
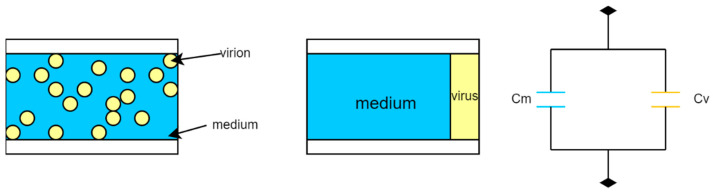
Electrical equivalent model of a two-zone parallel model distribution of virus particulate in a conductive medium, adapted from [23].

**Figure 6 micromachines-14-00345-f006:**
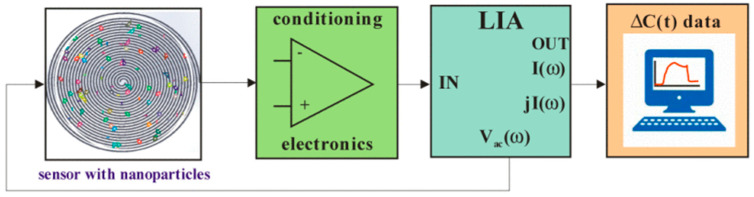
Capacitive-sensor based equipment to quantify the virus in a sample solution, from [39]. (Reproduced with permission from Guadarrama et al., IEEE International Conferences on Engineering in Veracruz; published by IEEE, 2020).

**Figure 7 micromachines-14-00345-f007:**
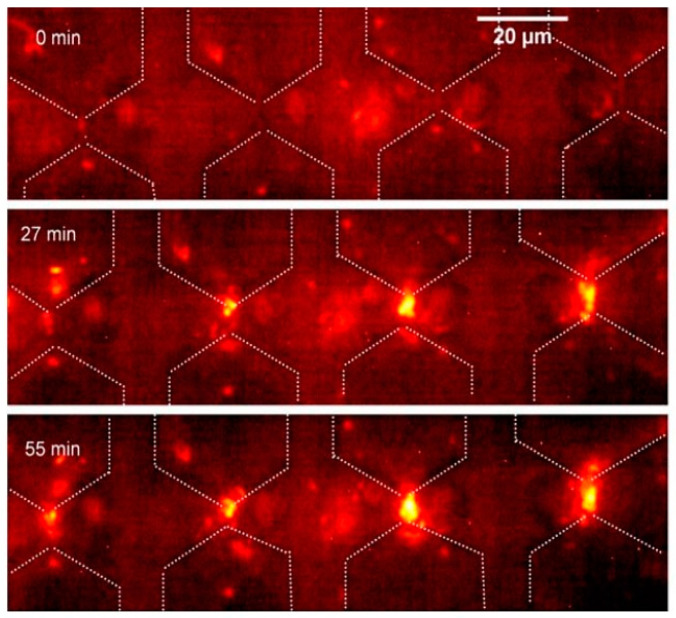
Observation of the lysis of the fluorescent stained Vaccinia virus under an SEM microscope, from [38]. Copyright obtained from the publisher.

**Figure 8 micromachines-14-00345-f008:**
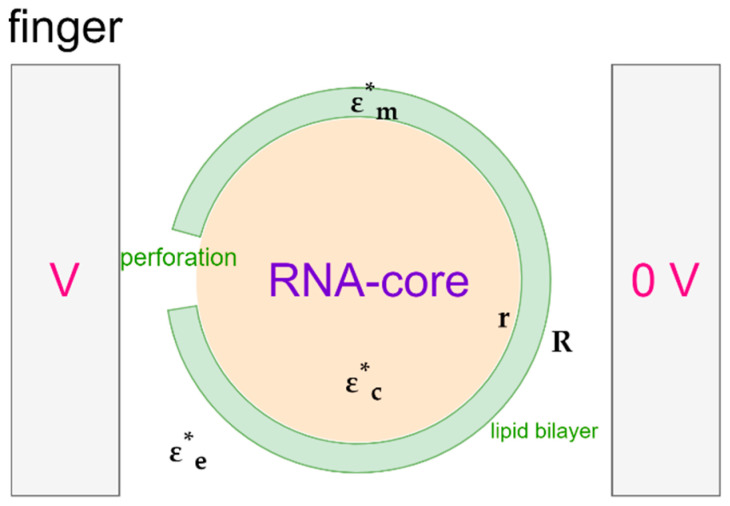
The electroporation of the membrane, a nonconductive lipid bilayer, can be achieved with DC or AC. The external medium, a buffer solution, and the core, can be expected to be conductive.

**Figure 9 micromachines-14-00345-f009:**
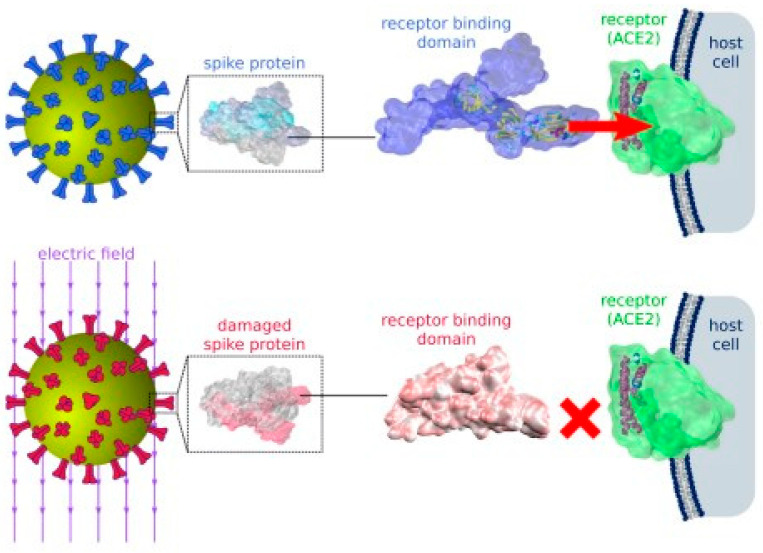
Inactivation of SARS-CoV-2 by damaging the spike protein using moderate electric fields, from [67]. (Reproduced with permission from Martin Garcia, Nature communications; published by *Nature*, 2021).

**Figure 10 micromachines-14-00345-f010:**
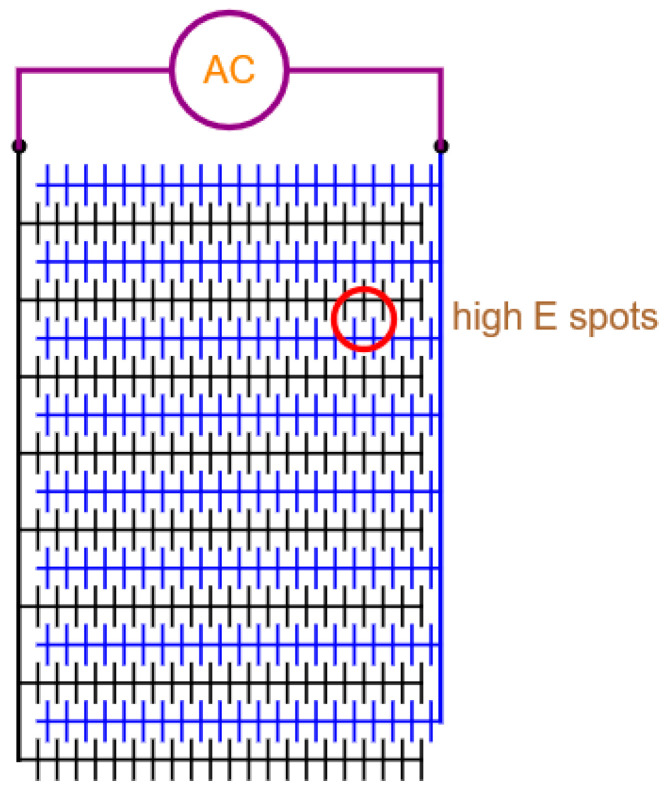
Planar interdigitated electrodes with spikes that are interlocked. In this way, many high electric field hotspots are created for inactivating viruses carried along in exhaled droplets or suspended in a buffer solution. The spacing between the fingers is 2 μm, and that between the tips of the spikes 500 nm.

**Table 1 micromachines-14-00345-t001:** Comparison of the practical studies that concentrated viruses using DEP.

Name	Size	Type	Capsulation	Trapping	Electrodes
Tobacco Mosaic	280 nm	RNA virus	Non-enveloped	pDEP	Sawtooth (6 µm) [13]
Herpes simplex	240 nm	DNA virus	Enveloped	pDEP	Quadrupole (6 µm) [15]
Vaccinia	360 × 270 × 250 nm	DNA virus	Enveloped	pDEP	Interdigitated (10 µm) [16]
Influenza	90 nm	RNA virus	Enveloped	nDEP	Quadrupole, Interdigitated (6 µm), (40 µm) [20]
Hepatitis A	27 nm	RNA virus	Non-enveloped	nDEP/pDEP	Quadrupole Octrupole (2 µm) [24]
Cowpea Mosaic	30 nm	RNA virus	Non-enveloped	pDEP	Castellated (2 µm) [21]
Adeno	90 nm	DNA virus	Non-enveloped	iDEP	Castellated/Interdigitated (10 µm) [22]
Sindbis	130 nm	RNA virus	Enveloped	iDEP	Sawtooth gradient (0–700 V) [26]
T4 bacteriophage	90 nm	DNA virus	Non-enveloped	iDEP	Circular and oval (80 µm) [27]

**Table 2 micromachines-14-00345-t002:** Comparison of the practical studies that sampled viruses using electric fields.

Virus Sampled	Virus Size	Sampling Mechanism	Sampling Time/Sampling Rate	Electrodes Used
Adenovirus, Rotavirus	90 nm, 70 nm	DEPIM (5 V_pp_, 100 kHz)	60 s	Castellated and interdigitated (10 µm) [22]
Vaccinia virus	360 × 270 × 250 nm	DEPIM (8 V_pp_, 1 kHz)	54 s 0.401 mm/s	Nanoelectrode array [35]
HIV, FIV	100 nm, 100 nm	Dopant concentration	<15 min	Co-axial resonator [36]
HIV, FIV, MPMV	100 nm, 100 nm, 250 nm	Capacitance measurement for electrically polarizable virus	NA	Co-axial resonator [23]
SDS micelle, copper nanoparticles	4 nm, 500 nm	Differential capacitance	200 s	Spiral electrode (120 µm) [39]
Influenza, TMV, Baculovirus	100 nm, 20 × 300 nm, 30 × 360 nm	DEPIM	Sampling time—a few minutes	Nano-gap electrodes (510 nm) [37]

**Table 3 micromachines-14-00345-t003:** A list of practical studies that inactivated biological samples using irreversible electroporation using electric fields.

Sample	Excitation	Results	Electrode Design
*Escherichia coli*	Pulsed excitation	Lysis observed at 3.5 V and 500 µs pulse	Spike electrodes [49]
*B. pertussis*	Pulsed excitation	Lysed with 300 V and 50 µs pulse	Matrix electrodes (15 µm) [51]
Leukemia, Red blood cells	DC biased AC excitation	Electrokinetic lysis reported due to forces caused by 145 V_rms_ and 1 kHz frequency across the microchannel	External electric field across a micro-channel [57]
*S. thermophilus*, *Escherichia coli*	AC excitation	Thermo-electric lysis reported caused by 240–280 V_rms_ and 20 kHz	Two electrophoresis electrodes and one lysis electrode [58]
Plant protoplast	AC excitation	Lysis observed at 10 V_pp_ and 10 MHz	Two electrodes across a trapezoid channel [59]
Vaccinia Virus	AC excitation	Lysis observed at 20 V_pp_ and 100 kHz	Spike electrodes [38]
Vaccinia virus	AC excitation	Electroporation observed at 8 V_pp_ and 1 kHz at a reduced flow velocity of 0.05 mm/s with a few particles irreversibly electroporated	Carbon nanoelectrode arrays [35]

## Data Availability

Not applicable.

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
