# Peer review of "Manipulation, Sampling and Inactivation of the SARS-CoV-2 Virus Using Nonuniform Electric Fields on Micro-Fabricated Platforms: A Review"

_micromachines, 2023, doi:10.3390/mi14020345_

Round 1
Reviewer 1 Report
* Congratulations on your work, which focuses on the possibility of using existing methods for manipulating and inactivating larger viruses and bacteria, on smaller viruses 15 like the SARS-CoV-2. The study is well-projected and the findings are fascinating. In order to ampliate the introduction section, adding some references to what is present in literature about this topic, I suggest you the reading of the following papers:
doi.org/10.1002/mco2.115; doi.org/10.1186/s43556-021-00033-4 ; doi.org/10.1186/s13287-022-02944-7.
* I suggest the authors incorporate some sentences of future perspectives related to the topic in the conclusion section.
* On page 8, line 242, the sentence "The differential capacitor due to the deposited nanoparticles is modeled as an effective homogeneous medium." needs the following reference:
doi.org/10.1016/j.ijhydene.2019.08.184; doi.org/10.3390/mi10010002
* Authors should place the full stop or comma after the reference number in the bracket [ ], not before it. There are also some formatting mistakes in the references section, I suggest the authors check and correct them.
* The English language needs to be improved with minor revisions.
Reviewer 2 Report
n this paper, authors have reviewed some electrical methods for the manipulation, concentration, detection, and inactivation of viruses, with a focus on the applications and prospects of these schemes for SARS-Cov-2. This review is interesting and well-organized and is suitable for publishing in Micromachines after minor revisions, some of which are mentioned below:
1- There are some typos in this article. For example:
In Keywords, the letter ‘m’ is bold.
In the second paragraph of page 2, ‘0’ should be written as a subscript for ‘ε0’.
In the last paragraph of page 1, the cross mark is omitted in ‘2.9 10E6’
Double-check for other possible typos.
2- In the third paragraph of the introduction, explain with more details and examples that how those electrical schemes can be used to develop other products, such as face masks and lab equipment. Provide some references for this purpose.
3- Please cite the previous review papers in this field and indicate the necessity of writing this review paper at the last paragraph of introduction section.
4- Provide some references and evidence for claims stated in the introduction. For example, for these claims: “… the COVID-19 pandemic, claimed over 6.37 million lives …” or “Optical methods to quantify viruses often need expensive equipment …”
5- At the beginning of Section 2, the authors have defined the term “Dielectrophoresis”. Explain what “negative dielectrophoresis” and “positive dielectrophoresis” are.
6- In the text, refer to equations and figures in their correct form, according to the micromachines template. For example, rewrite “eq. (1)” into “Equation (1)”, or “Fig.3” into Figure 3”.
7- Refer to all figures in the text before bringing the figure. For example, the authors have not mentioned and discussed Figure 1 in the text at all, or have mentioned Figure 2 in the next paragraphs after providing the figure.
8- Some disorders in citations can be seen. For example, ref [76] is cited between refs [22] and [23].
9-Figures 5 and 9 have interrupted the text and are pasted in the middle of the text. Replace these figures to an appropriate location so that they do not cut the paragraph.
10- The quality of the figures should be improved. Also, if some Figures are borrowed from other articles, in addition to citing that article, the publisher’s permission are required. The authors should state publisher permission in the figure caption. (See Micromachines’ policies for this case)
11- The authors should collect the data of papers in a table so that the reader can compare the different aspects of the papers mentioned in this review paper all in one.
12- It is recommended that the conclusion section does not contain any citations. Instead of referring to Arbeitman’s work, explain briefly and restate their postulates. Other future directions should also be stated.
Round 2
Reviewer 2 Report
The authors declined to add a table to collect the data so that the reader can compare the different aspects of the papers mentioned in this review paper all in one. I recommended adding three tables for 3 sections, including concentration, sampling, and inactivation.
Author Response
The tables covering all the practical studies on the concentration, sampling, and inactivation of viruses have been added.
Round 3
Reviewer 2 Report
The authors have revised the manuscript based on the reviewers' comments and the manuscript can be published in the current format.